# Integrated 4D Analysis of Intramuscular Fat Deposition: Quantitative Proteomic and Transcriptomic Studies in Wannanhua Pig Longissimus Dorsi Muscle

**DOI:** 10.3390/ani14010167

**Published:** 2024-01-04

**Authors:** Xiaojin Li, Fei Xie, Ruidong Li, Lei Li, Man Ren, Mengmeng Jin, Ju Zhou, Chonglong Wang, Shenghe Li

**Affiliations:** 1College of Animal Science, Anhui Science and Technology University, Chuzhou 239000, China; lixj@ahstu.edu.cn (X.L.); xiefei19980430@126.com (F.X.); m17563400413@163.com (R.L.); lil@ahstu.edu.cn (L.L.); renm@ahstu.edu.cn (M.R.); jinmm@ahstu.edu.cn (M.J.); 2Anhui Province Key Laboratory of Animal Nutritional Regulation and Health, Chuzhou 233100, China; 3Institute of Animal Husbandry and Veterinary Medicine, Anhui Academy of Agricultural Sciences, Hefei 230031, China; 4Kunshan Animal Health Supervision Institute, Kunshan 215300, China; zhouju0227@126.com

**Keywords:** Wannanhua pig, proteomics, IMF, transcriptomics, longissimus dorsi muscle, PRM

## Abstract

**Simple Summary:**

Wannanhua is a pig breed indigenous to Anhui Province, China. This breed has a high intramuscular fat content, making it an ideal model for investigating lipid deposition mechanisms in pigs. Intramuscular fat content is one of the main indicators of meat quality in pigs; however, the molecular mechanisms underlying intramuscular fat deposition in skeletal muscle remain unknown. We conducted a comprehensive proteomic and transcriptomic analysis of Wannanhua pig longissimus dorsi muscle tissues to identify the genes, proteins, and molecular pathways involved in intramuscular fat deposition. By integrating transcriptomic data, we identified seven candidate genes including ACADL, ACADM, ANKRD2, MYOZ2, TNNI1, UCHL1, and ART3 that play a regulatory role in fat deposition and muscle development. These findings establish a theoretical foundation for future analyses of lipid deposition traits, contributing to potential enhancements in pig meat quality during breeding and advancing the selection process for Chinese indigenous breeds.

**Abstract:**

Wannanhua (WH) is a pig breed indigenous to Anhui Province, China. This breed has a high intramuscular fat (IMF) content, making it an ideal model for investigating lipid deposition mechanisms in pigs. IMF content is one of the main indicators of meat quality in pigs and is regulated by multiple genes and metabolic pathways. Building upon our prior transcriptomic investigation, the present study focused on the longissimus dorsi muscle tissue of Wannanhua (WH) pigs in the rapid fat-deposition stages (120 and 240 days of age). Employing 4D label-free quantitative proteomic analysis, we identified 106 differentially expressed proteins (DEPs). Parallel reaction monitoring (PRM) technology was used to verify the DEPs, and the results showed that the 4D label-free results were reliable and valid. Functional enrichment and protein–protein interaction analyses showed that the DEPs were mainly involved in the skeletal-muscle-associated structural proteins, mitochondria, energy metabolism, and fatty acid metabolism. By integrating transcriptomic data, we identified seven candidate genes including ACADL, ACADM, ANKRD2, MYOZ2, TNNI1, UCHL1, and ART3 that play a regulatory role in fat deposition and muscle development. These findings establish a theoretical foundation for future analyses of lipid deposition traits, contributing to potential enhancements in pig meat quality during breeding and advancing the selection process for Chinese indigenous breeds.

## 1. Introduction

Muscle growth and intramuscular fat (IMF) deposition are economically important traits affecting the meat quality and production performance of pigs [1,2,3]. IMF content is closely related to the flavor, tenderness, and juiciness of pork, a major source of protein among humans. In addition to ideal taste, appropriate IMF content determines the quality of meat [4]. The Wannanhua (WH) pig is an excellent indigenous pig breed found in Anhui Province, China. The marbling and IMF content of these pigs are significantly higher than those of Western commercial pig breeds (*p* < 0.01) [5,6]. Therefore, the WH pig is an ideal model for studying lipid deposition to elucidate the molecular mechanisms related to muscle growth and fat deposition in pigs, and to provide a theoretical basis for further screening of molecular markers related to muscle growth and fat deposition in the later stage.

Our study of WH pigs at different developmental stages revealed the fastest muscle development and fat deposition from 120 days to 240 days. In addition, differentially expressed genes (DEGs) involved in lipid biosynthesis were identified [7]. However, a major limitation of transcriptomics is that mRNA expression levels and actual protein abundance cannot be directly correlated. Different cells or tissues express varying amounts and types of proteins. Furthermore, the same cells or tissues express different amounts and types of proteins at varying stages of development. Dynamic changes in the abundance of proteomes within cells are essential for various life processes. Quantitative proteomics is the precise quantification and identification of all the proteins expressed in a genome or in a complex system. Therefore, proteomic studies are needed to support transcriptomic results and more accurately interpret changes in biological protein expression patterns.

With the rapid development of high-throughput sequencing technology in recent years, multi-omics studies such as transcriptomics and proteomics have considerably facilitated the elucidation of molecular mechanisms underlying IMF deposition [8,9]. Proteomic approaches based on two-dimensional gel electrophoresis and mass spectrometry (MS) have been used to study porcine skeletal muscle. However, the mechanisms underlying adipogenesis and regulation remain unclear [10,11]. Label-free quantitative proteomic analysis with higher sensitivity (to detect low-abundance proteins), higher reproducibility, and more reliable peptide identification and quantitative analysis enables the identification and differential expression analysis of almost all protein types [12,13].

In this study, we utilized transcriptome and 4D-Label-free quantitative proteome-based proteomic analyses to investigate differences in transcriptome and protein profiles of 120 and 240 days longissimus dorsi (LD) tissue samples from WH pigs. This study aimed to identify candidate genes and key pathways of lipid deposition, thereby providing a theoretical basis for further studies of lipid regulatory networks and molecular mechanisms underlying the postnatal development of porcine skeletal muscle. This could help improve pork quality and promote the growth of Chinese indigenous pig breeds, especially WH pigs.

## 2. Materials and Methods

### 2.1. Ethics Statement and Collection of Tissue Samples

The six castrated male pigs used in this experiment were all from a WH pig breeding farm in Huangshan City, China. The slaughtering and sampling experiments were conducted in accordance with the guidelines for the care and use of experimental animals. Experiments were performed according to the Regulations for the Administration of Affairs Concerning Experimental Animals and approved by the Animal Research Committee of Anhui Science and Technology University (approval number: 2021-316; date of approval: November 2021).

Pigs were reared under the same environmental and nutritional conditions. LD muscle samples from WH pigs at two developmental stages (120 days and 240 days) were collected for proteomic analysis of three biological replicates, with at least three LD muscle samples collected for each replicate. Three individual WH pigs at the same stage of development had similar weights. Commercial feed and water were offered ad libitum to all pigs. A section of the LD muscle close to the third or fourth lowest rib was manually dissected from each pig immediately after euthanizing. The samples were immediately frozen, placed in a liquid nitrogen tank, and transferred to a laboratory freezer at −80 °C for storage.

### 2.2. Protein Extraction, Trypsin Digestion

Appropriate amounts of LD muscle samples were pre-cooled with liquid nitrogen, placed into a mortar, and ground to powder. The powder was transferred to clean 5 mL tubes, and cracking buffer (8 mol/L Urea, 1% Protease inhibitor) 4× the volume of the sample was added. After ultrasonic cracking, the powder was centrifuged at 4 °C for 10 min at 12,000 rpm. The supernatant was collected, and the protein concentration was determined using the BCA kit (Thermo Fisher Scientific, Waltham, MA, USA).

Dithiothreitol solution was added to the protein solution until the protein concentration reached 5 mmol/L. The solution was subsequently reduced at 56 °C for 30 min. Next, 100 mmol/L iodoacetamide solution was added until the final protein concentration reached 11 mmol/L, and the solution was stored at room temperature for 12 min and away from light. The concentration of urea in the protein sample solution was diluted to <2 mol/L using tetraethyl ammonium bromide solution to eliminate the effect of urea on pancreatic enzyme activity. Pancreatic enzyme was added at 2% of the protein mass. Following enzymatic hydrolysis at 37 °C for 12 h, pancreatic enzyme was added at 1% of the protein mass. Then, enzymatic hydrolysis was performed at 37 °C for 4 h. A C18 solid phase extraction column was used to extract the peptide after enzymolysis.

### 2.3. LC-MS/MS Acquisition and Database Search

According to the quantitative results, the 2 μg enzymatic hydrolysis product was analyzed by LC-MS/MS. Separation was performed using the NanoElute system (Bruker, Bremen, Germany). The samples were fed into IonOpticks (Australia, 25 cm × 75 μm, C18 packing 1.6 μm) by an automatic injector and separated at a flow rate of 300 nL/min. The samples were separated by chromatography and then analyzed by mass spectrometer with timsTOF Pro (Bruker, Bremen, Germany). The parent ion of peptide segment and its secondary fragments were detected and analyzed by TOF. The secondary mass spectrometry scan range was set to 100–1700 *m*/*z*. PASEF mode was used for data acquisition. After primary mass spectrometry collection, secondary spectra with charge number of parent ions in the range of 0–5 were collected in PASEF mode 10 times, and the dynamic exclusion time of serial mass spectrometry scanning was set to 30 s to avoid repeated scanning of parent ions.

Secondary mass spectrometry data were retrieved using Maxquan software 1.6.14.0 (Thermo Fisher Scientific, Lenexa, KS, USA). The LC-MS/MS data were searched for in the “Uniprot_Sus scrofa (Pig) 329836_20220314_9823” database (http://www.uniprot.org (accessed on 15 November 2022)). After passing quality control, the final reliable protein identification result was obtained according to the set threshold value. Finally, a quantitative analysis was conducted.

### 2.4. Protein Identification and Bioinformatic Analyses

Proteomic data were first analyzed using the Statistical Program for Social Sciences (SPSS) (SPSS Inc., version 20.0, Chicago, IL, USA). The significant DEPs were screened with a fold change ≥1.5 or ≤0.67 and *p*-value ≤ 0.05 using a t-test. Next, a multiple testing correction was performed via the Benjamini and Hochberg procedure to control the FDR using *p*-value < 0.05. Gene Ontology (GO) and Kyoto Encyclopedia of Genes and Genomes (KEGG) analyses were used to determine the roles of DEPs. Proteins with *p*-value <0.05 were considered significantly enriched. Details of GO and KEGG assays are described in our previous study [5].

### 2.5. Parallel Reaction Monitoring (PRM) Verified Typical Specific Expression Proteins

Samples from the LD120 and LD240 groups were separated by the NanoElute system with nanoliter flow rate and then injected into the CaptiveSpray ion source for ionization and analysis by timsTOF Pro (Bruker, Bremen, Germany) mass spectrometer. The data acquisition mode was PRM-PASEF. The primary mass spectrometry automatic gain control (AGC) and maximum ion implantation time (maximum IT) were set to 3 × 10^6^ and 50 ms, respectively. The secondary mass spectrometry AGC, maximum IT, and isolation window were set to 1 × 10^5^, 100 ms, and 2.0 *m*/*z*, respectively.

### 2.6. Correlation Analysis of Transcriptomics and Proteomics

Correlating transcriptome sequencing and proteomic data provide a deeper understanding of gene transcription and post-transcriptional regulatory mechanisms. In this study, association analyses included gene and protein expression, GO, and KEGG association analyses. The screening condition for significant differences was *p*-value < 0.05.

### 2.7. Statistical Analyses

SPSS 22.0 software was used for statistical analysis of the data, and a non-parametric test was selected according to whether the data conformed to the normal distribution. One-way ANOVA was used to analyze variance. The results were expressed as the mean ± SE, and a *p*-value < 0.05 was considered statistically significant.

## 3. Results

### 3.1. Characterization of the Identified Proteins

The LD muscle samples obtained from WH pigs of different ages (LD120 and LD240) were quantitatively analyzed through 4D Label-free proteomics, and 464,461 secondary spectra were identified through a database search. Of these, 105,678 spectra matched the reference database, with 15,942 and 1923 peptides and proteins matched, respectively (Figure 1A). The length distribution of the identified peptides are shown in Figure 1B. Most of the peptides ranged from 6 to 18 amino acids in length; proteins with 13 amino acids were the most abundant. The number of proteins identified by a different number of peptide segments is shown in Figure 1C. Most of the proteins were identified by 1–20 peptide segments. Among these, up to 271 proteins contained two peptide segments (Figure 1C). The molecular weights of the identified proteins were widely distributed, with most ranging from 10–80 kDa (Figure 1D).

### 3.2. Protein Identification and Differential Expression

A total of 1150 proteins were identified. Of these, 106 DEPs (45 upregulated and 61 downregulated) were identified between the LD120 and LD240 groups (Figure 2A). The hierarchical clustering method was used to analyze the abundance changes of the screened differential proteins in the LD muscle of WH pigs at different ages, and the results are shown in Figure 3. There were significant differences in protein expression within the LD muscle tissue between the LD240 (L240-1, L240-2, L240-3) and LD120 (L120-1, L120-2, L120-3) groups. Subcellular location analysis revealed that the DEPs were mainly cytosolic (33.5%), extracellular (22.0%), nucleic (16.8%), and mitochondrial (15.6%) (Figure 2B).

### 3.3. Bioinformatic Analyses

The DAVID version6.8 software was used to initially explore the potential functions of the identified DEPs in silicosis. These DEPs were enriched in 1606 GO terms (Figure 4). The BP category was associated with the regulation of cytoskeleton organization, positive regulation of phospholipid biosynthetic processes, and regulation of skeletal muscle contraction via the regulation of sequestered calcium ion release. The CC category was associated with the fibrinogen complex, myosin filament, and myosin phosphatase complex. The MF category was associated with JUN kinase binding, G protein-coupled receptor activity, and myosin phosphatase activity. KEGG pathway analysis indicated that DEPs in both groups were involved in 25 pathways (Figure 5), including PPAR signaling, vascular smooth muscle contraction, fatty acid metabolism, the PI3K-Akt signaling pathway, fatty acid degradation, and neuroactive ligand–receptor interaction.

### 3.4. Protein–Protein Interaction (PPI) Network Analysis

A network of physical and functional PPI was constructed using the CytoScape v.3.2.1 online software against the Sus scrofa database. Differential proteins were closely related, and their functional modules were mainly the PPAR signaling pathway, fatty acid metabolism, and the PI3K-Akt signaling pathway. In addition, these three pathways overlapped with KEGG annotation results, suggesting that they play an important role in fat deposition in the LD muscle of WH pigs at the ages of 120–240 days. The predicted protein interaction network comprised 23 and 22 proteins whose expression was upregulated and downregulated, respectively (Figure 6). The predicted PPI network of DEPs revealed that APOH, PCYOX1, CPT2, MYH7, ACADM, ACADL, DBI, PARK7, RPS20, RPS12, and RPL4 have central roles.

### 3.5. PRM Validation of Protein Expression

Ten DEPs were selected for PRM verification based on the proteomic analysis, namely UQCRB, DDT, NDUFA7, PFDN5, CKM, GAPDH, TUBA1A, PHPT1, TNNC1, and ATP5ME (Figure 7). These DEPs are involved in fat metabolism, muscle growth and development, and the PPAR signaling pathway. The PRM verification results for these proteins were consistent with those of the 4D Label-free analysis, suggesting that 4D Label-free proteomic analysis is reliable.

### 3.6. Comparative Analysis of Proteomic and Transcriptomic Data

The mRNA information obtained from the transcriptome and protein information identified via the proteome were integrated and analyzed. All DEGs and DEPs were divided into nine quadrants according to the significant difference thresholds. The nine-quadrant map is shown in Figure 8 (Pearson’s correlation coefficient, R = 0.54, *p*-value = 2.2 × 10^−16^). In the fifth quadrant, mRNA and protein were not differentially expressed. In Quadrants 3 and 7, mRNA and protein had consistent differential expression patterns. In contrast, mRNA and protein had inconsistent differential expression patterns in the remaining quadrants, with potential post-transcriptional or translational level regulation. The data points in Quadrants 3 and 7 revealed an overlap in nine genes between DEGs and DEPs. Of these, the expression of seven genes was upregulated (ACADL, ACADM, ANKRD2, LRRC20, MYOZ2, TNNI1, and UCHL1), whereas that of two genes was downregulated (ART3 and SERPINA1). These cor-DEGs-DEPs genes might play important roles in muscle development and fat deposition.

To further explore the potential functions of the cor-DEGs-DEPs genes in muscle development and fat deposition, GO terms and KEGG pathways were enriched at the transcriptomic and proteomic levels, respectively. GO terms were highly enriched at both the mRNA and protein levels, including the regulation of cytoskeleton organization, lipid metabolic processes, ATP metabolic processes, the cytokine-mediated signaling pathway, and fatty acid metabolic processes (Figure 9). KEGG pathway analysis indicated that 31 pathways were highly enriched at both the mRNA and protein levels, including the PPAR, Fc epsilon RI, Phospholipase D, and NF-kappa B signaling pathways, as well as vascular smooth muscle contraction (Figure 10).

## 4. Discussion

Muscle growth and IMF content in pigs are important economic traits. Skeletal muscle development and IMF deposition are regulated by multiple genes and metabolic pathways. In our previous study [5], we measured the dynamic changes of IMF content at three developmental stages (60 days, 120 days, and 240 days), demonstrating that IMF deposition in the LD muscle of WH pigs was the fastest stage from 120 to 240 days (*p*-value < 0.01). In addition, transcriptomic changes in the LD muscle at 120 days and 240 days in WH pigs were analyzed to identify DEGs related to muscle growth and development and lipid metabolism. However, the main steps of gene expression include transcription and protein synthesis, with mRNA as the intermediate of gene expression and protein as the executive of gene function. In order to understand the regulation process of muscle development and fat-deposition-related gene expression at the fastest stage of IMF deposition in WH pigs, the simultaneous monitoring of mRNA and protein is required. The differences between and complementarity of transcriptomics and proteomics to measure gene expression levels are helpful for fully understanding the functions of these genes at the most rapid stage of IMF deposition and for analyzing the muscle development and fat deposition regulatory mechanisms in the LD muscle.

In the present study, using proteomic analyses based on 4D Label-free technology, 106 DEPs were identified between the LD120 and LD240 groups at different developmental stages of LD muscle in WH pigs. Among these, the expression of 45 and 61 proteins was upregulated and downregulated, respectively (Figure 2A). The upregulated proteins that differed most significantly in the LD240 group were MYOZ2 and UCHL1 (Appendix A). *MYOZ2* encodes calcineurin Myozenin2 (MYOZ2), a muscle tissue-specific protein that is specifically expressed in slow skeletal muscle fibers and regulates muscle growth mainly through the calcineurin-NFAT signaling pathway. *Myoz2* is an important candidate gene for pig growth traits [14,15]. Similarly, UCHL1 is a novel regulator of skeletal muscle mitochondrial function and oxidative activity. It modulates lipid metabolism, mainly by stabilizing the adipocyte differentiation-associated protein (PLIN2) [16,17]. The most significantly downregulated proteins in the LD240 group were ART3 and SERPINA3 (Appendix A). ART3, a member of the mono-ADP-ribotransferase family, can affect cell division and differentiation by participating in biological processes, such as DNA repair, signal transduction, and metabolic cascade [18]. Previous studies have shown that ART3 expression can induce phosphorylation of extracellular regulated protein kinase (ERK) and serine/threonine kinase Akt (AKT) [19]. ERK belongs to the mitogen-activated protein kinase (MAPK) family. The MAPK signaling pathway is a bridge between extracellular stimuli (including growth factors and hormones) and intracellular gene expression regulation and can participate in the regulation of adipocyte differentiation and skeletal muscle development [20,21]. AKT is involved in a variety of cellular processes, including proliferation, metabolism, and cell size regulation [22]. The PI3K/AKT signaling pathway is involved in the regulation of important biological processes, such as muscle growth and development, metabolic regulation, and homeostasis maintenance [23,24,25]. Therefore, it is speculated that ART3 protein may regulate muscle growth and development and lipid metabolism by mediating MAPK and PI3K/AKT signaling pathways. SERPINA3 belongs to the serine protease inhibitor family [26]. SERPINA3 protein is regulated by both insulin and growth factors, which affects cell proliferation and can promote the growth and development of skeletal muscle [27,28]. In this study, the expression of ART3 and SERPINA3 in the LD120 group was significantly higher than that in the LD240 group. This could be because these proteins regulate cell metabolism by modulating mitochondrial function, thereby affecting tissue development and fat deposition in the LD muscle of WH pigs. As a result, the IMF content in LD muscle at the age of 120 days was significantly lower than that at 240 days. This is consistent with the results of the previous study which measured the intramuscular fat content of LD muscle at different ages [7].

GO annotation results indicate that CARNS1 and SLC25A6 are closely related to muscle development and metabolism. KEGG annotation and protein network interaction analyses revealed that different proteins were mainly enriched in the PPAR signaling pathway, fatty acid metabolism, and PI3K-Akt signaling pathway at 120 and 240 days of age. Fatty acid metabolism is a complex process in which some key proteins play an important role. The high expression of PPAR signaling pathway-related proteins can lead to an increase in lipid metabolism, which is consistent with previous reports [29]. In addition, these results are consistent with the slow growth rate, high intramuscular fat deposition capacity, and better meat quality of Chinese indigenous pig breeds, such as WH pigs. PPI analysis was performed for differential proteins associated with fat deposition and muscle development. It was found that ACADL, ACADM, and CPT2 were closely related, and all of them were at important PPI nodes. Long-chain acyl-CoA dehydrogenase (ACADL) is involved in the β-oxidation of long-chain fatty acids [30,31]. Medium-chain acyl-CoA dehydrogenase (ACADM) is involved in the β-oxidation process of medium-chain fatty acids and amino acids [32]. CPT2 is a key rate-limiting enzyme for the β-oxidation of long-chain fatty acids [33]. β-oxidation is the main process of fatty acid oxidation, in which fatty acids are activated to generate fatty acyl-CoA; the activated fatty acyl-CoA enters the mitochondria and is oxidized, being the main source of biological energy for the body or tissues [34]. Studies have shown that ACADL, ACADM, and CPT2 participate in fatty acid metabolism in porcine skeletal muscle and are associated with IMF deposition and muscle development in pigs [35,36]. In the present study, the expression levels of ACADL, ACADM, and CPT2 in the LD240 group were significantly higher than those in the LD120 group. These results indicate that these proteins are related and can regulate muscle growth and fat deposition through interaction, which may be the cause of the high-fat deposition ability of WH pigs in the LD240 group.

In this study, we comprehensively analyzed the transcriptomic and proteomic data of the LD muscle of WH pigs. Only differences in mRNA and protein levels of ACADL, ACADM, ANKRD2, LRRC20, MYOZ2, TNNI1, UCHL1, ART3, and SERPINA1 were found between the LD120 and LD240 groups. Several more genes were expressed inconsistently at the transcriptional and protein levels. This is consistent with previous findings [37,38], indicating that the regulation of genes from transcription to translation cannot be ignored. TNNI1 belongs to the multi-gene family of calcium ion-binding proteins. This protein exists in different types of skeletal muscle and mainly affects muscle development and IMF deposition. Furthermore, the increase of IMF content is positively correlated with increased TNNI1 abundance [39,40]. This is consistent with the results of this study, which demonstrated that the expression level of TNNI1 in the LD240 group was significantly higher than that in the LD120 group. ANKRD2 is a member of the muscle anchor–protein repeat protein family and plays an important role in porcine skeletal muscle growth [41]. In summary, *ACADL*, *ACADM*, *ANKRD2*, *MYOZ2*, *TNNI1*, *UCHL1*, and *ART3* are all candidate genes that affect IMF deposition and muscle development. In future work, we will focus on the specific mechanisms of cor-DEG-DEP genes involved in regulating IMF deposition and muscle development in pigs.

## 5. Conclusions

In summary, we used the 4D Label-free proteomic method to perform differential proteomic analysis of the stages (120 days and 240 days) at which IMF deposition rates were fastest in the LD muscle of WH pigs. We identified 106 DEPs, which enhanced our understanding of the differences in fat deposition at the protein level. After combining this proteomic data with previous transcriptomic data, seven genes were identified as being associated with fat deposition and muscle growth. These genes were divided into lipid-deposition-related genes (*ART3*), lipid metabolism regulators centered on PPAR signaling pathways (*ACADM* and *ACADL*), and muscle growth and development related genes (*ANKRD2*, *MYOZ2*, *UCHL1*, and *TNNI1*). According to the results of this study, the molecular genetic mechanism of fat deposition during the growth and development of WH pigs is complex and determined by multiple genes. In conclusion, our study provides valuable information for the further study of the molecular mechanisms of muscle growth and lipid deposition traits in pigs, especially in WH pigs, which may help in the genetic improvement of meat quality traits.

## Figures and Tables

**Figure 1 animals-14-00167-f001:**
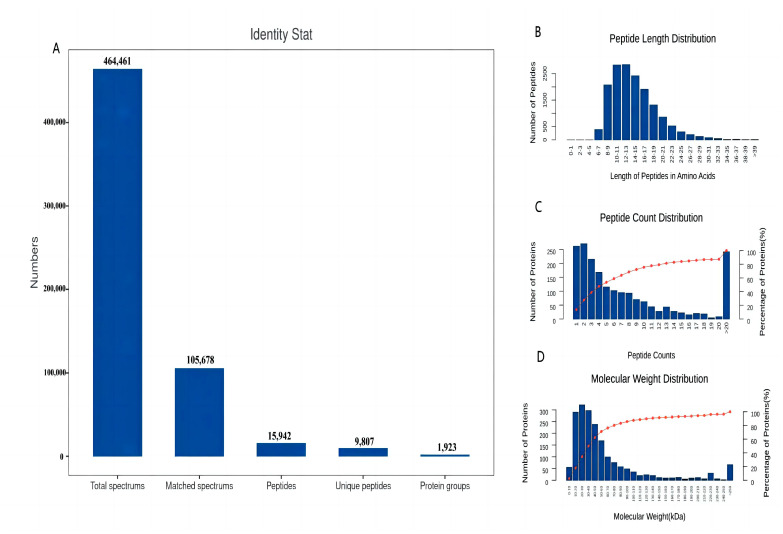
Characterization of proteins from six longissimus dorsi muscle samples. (**A**) Overview information of identified proteins in this study. (**B**) Distribution of the lengths of the identified peptides. (**C**) Distribution of the numbers of identified proteins containing different numbers of peptides. (**D**) Distribution of the molecular weights of the identified proteins.

**Figure 2 animals-14-00167-f002:**
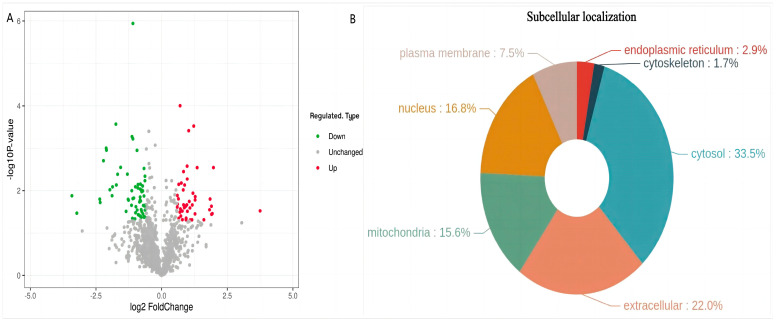
Differentially expressed proteins (DEPs) between the LD120 and LD240 groups. (**A**) Volcano plot of proteins in the LD240 and LD120 groups. (**B**) Subcellular localization chart of DEPs.

**Figure 3 animals-14-00167-f003:**
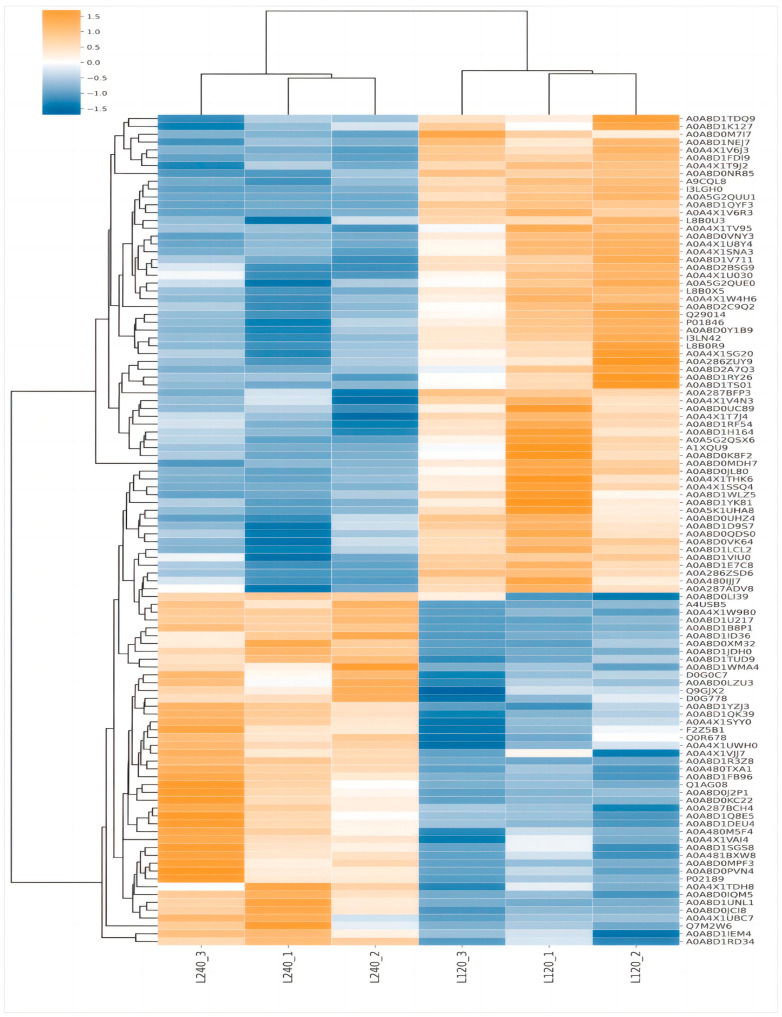
The hierarchical cluster analysis of DEPs between the LD120 and LD240 groups.

**Figure 4 animals-14-00167-f004:**
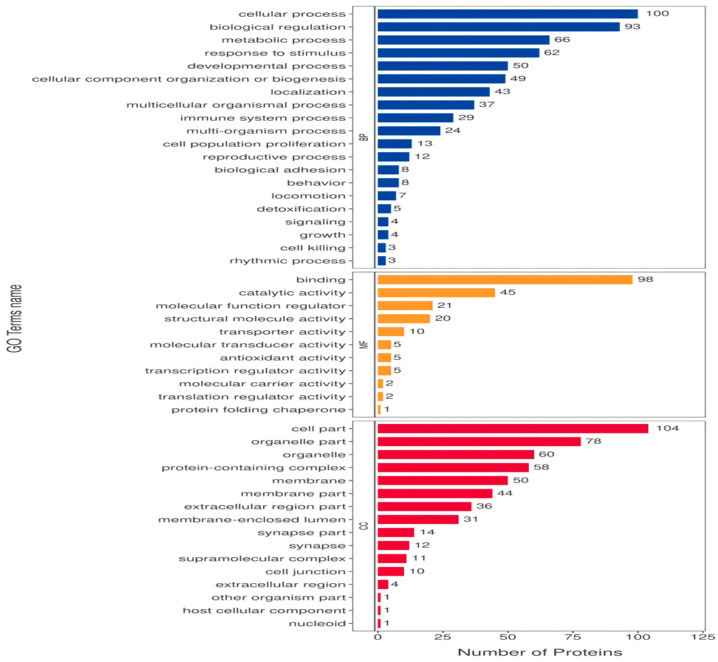
The GO secondary annotation of DEPs between the LD120 and LD240 groups.

**Figure 5 animals-14-00167-f005:**
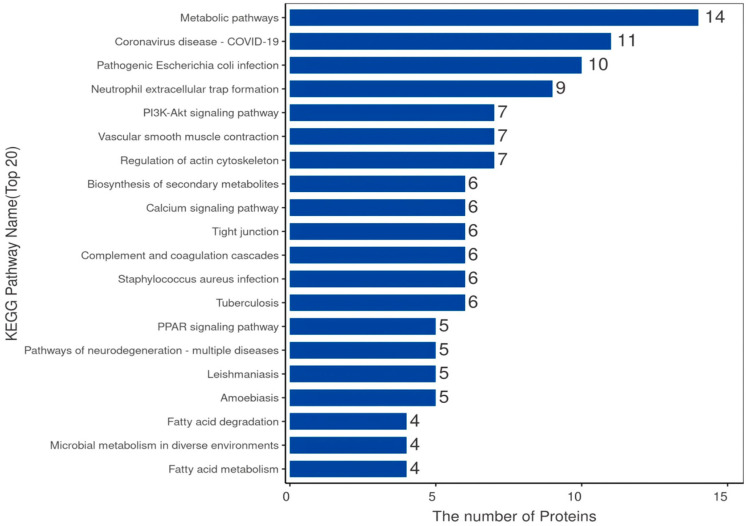
Scatter diagram of the enriched KEGG pathways. The top 20 terms are shown. The Y-axis represents the name of the pathway, and the X-axis represents the number of proteins.

**Figure 6 animals-14-00167-f006:**
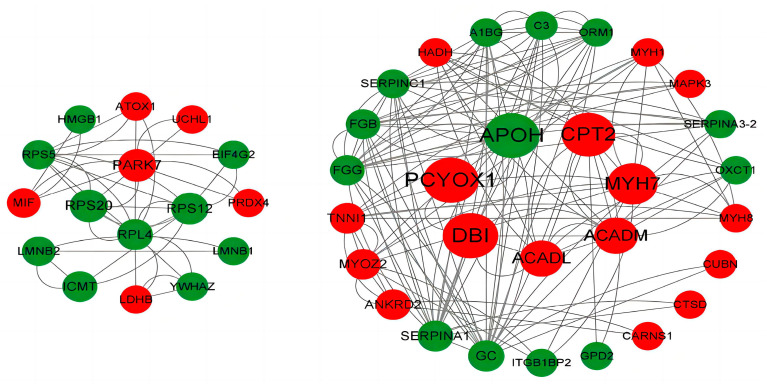
Protein–protein interaction (PPI) network of DEPs between the LD120 and LD240 groups. Upregulated proteins are presented in red and downregulated proteins are presented in green. Node size indicates the value of node betweenness; larger nodes are more important in PPI network stability. The higher gray value of the edge indicates a higher protein–protein interaction score.

**Figure 7 animals-14-00167-f007:**
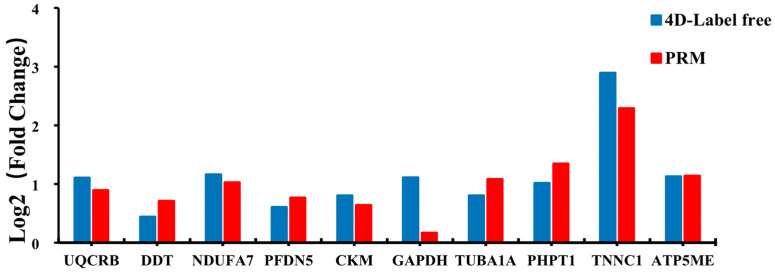
Protein relative expression values in the parallel reaction monitoring (PRM) experiment.

**Figure 8 animals-14-00167-f008:**
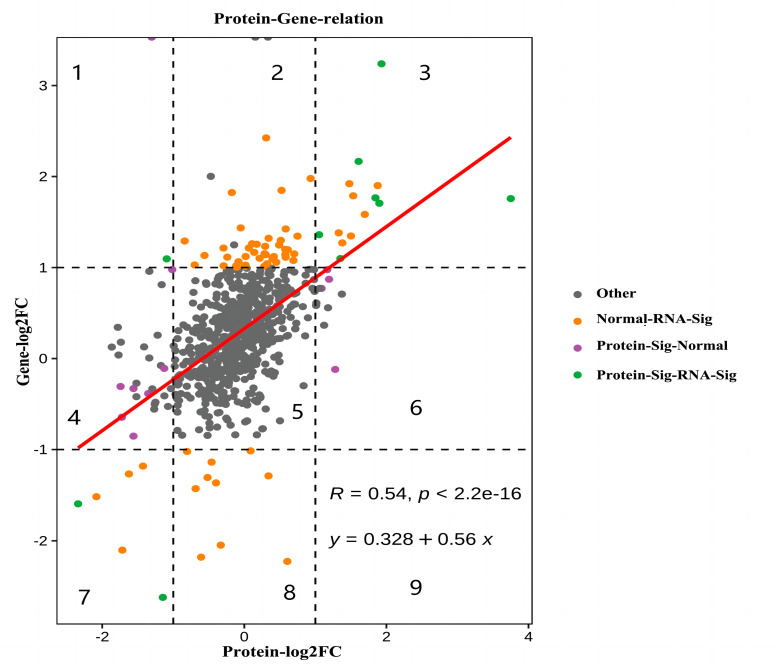
Correlation analysis of DEPs–DEGs in WH pigs at different developmental time points.

**Figure 9 animals-14-00167-f009:**
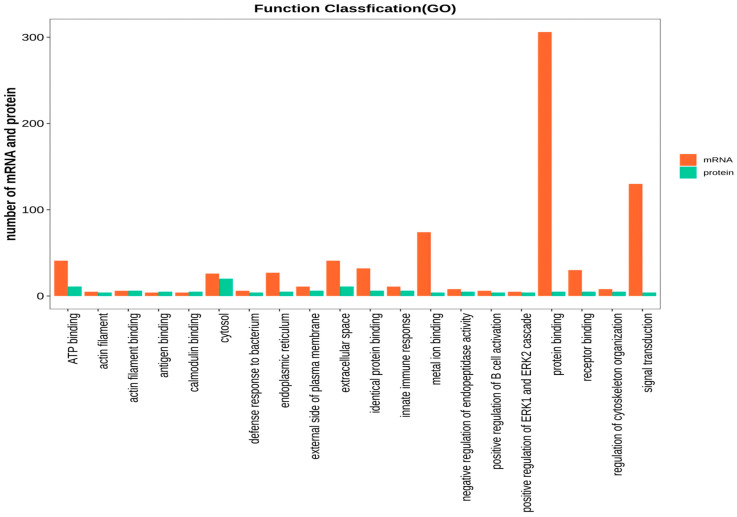
Correlation of GO enrichment between the transcriptome and proteome.

**Figure 10 animals-14-00167-f010:**
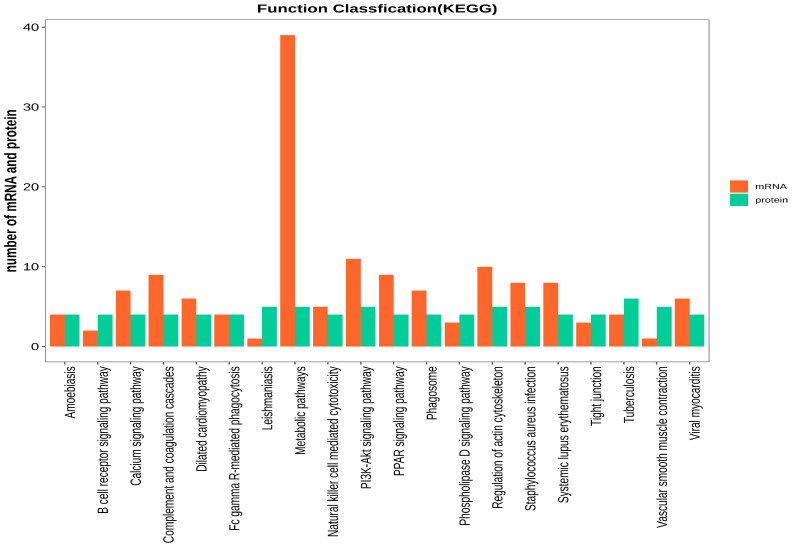
Correlation of KEGG enrichment between the transcriptome and proteome.

## Data Availability

The transcriptomics datasets generated during the current study are available in NCBI SRA (PRJNA915318) and the mass spectrometry proteomics data have been deposited to the ProteomeXchange Consortium via the PRIDE partner repository with the dataset identifier PXD046675.

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
