# Peer review of "Integrated 4D Analysis of Intramuscular Fat Deposition: Quantitative Proteomic and Transcriptomic Studies in Wannanhua Pig Longissimus Dorsi Muscle"

_animals, 2024, doi:10.3390/ani14010167_

Round 1

Reviewer 1 Report

Comments and Suggestions for Authors

Attached.

Author Response

  1. There is evidence to show the mechanism underlying intramuscular fat deposition. I disagreewith the title of the paper. This study is just a correlation study.

Reply: Thank you very much for your suggestion. It has been revised in the newly submitted paper.

  1. Transcriptomic data is used, having already been published in the reference listed here. All ofthe genes identified in this paper differ from the current study. There is no explanation for why proteomic and transcriptomic data are so different.

Reply: Thank you very much for your suggestion. Proteomics is the quantitative and qualitative analysis of all proteins in cells, tissues, organs and organisms at the whole level. Proteomic studies can assist transcriptomic results and more accurately analyze the changes in protein expression patterns of organisms. The present study was based on the transcriptomic analysis of the fastest stage of IMF deposition (LD120 and LD240 ) in WH pigs.

  1. Line 289- LD24 should be changed to LD240.

Reply: Thank you very much for your suggestion. It has been revised in the newly submitted paper.

  1. The gender of the animal is not mentioned in the methods section.

Reply: Thank you very much for your suggestion. It has been revised in the newly submitted paper.

  1. Other methodologies should validate the target gene identified in this study to ensure adifferential expression in this gene between the LD120 and LD240 groups. Other than based on pathways analysis, there is no evidence to suggest that these genes are involved in fat deposition.

Reply: Thank you very much for your suggestion. Parallel response monitoring (PRM) is a targeted proteomic technique and an effective method to validate targeted proteins. In this study, PRM technique was used to verify the screened differential proteins.

  1. Is there a difference in fat content between LD120-LD240?

Reply: Thank you very much for your suggestion.There were significant differences in IMF content between the two groups. The measurements have been published in a previous transcriptome data paper[Li X et al., 2023].

Li X, Yang Y, Li L, Ren M, Zhou M, Li S. Transcriptome Profiling of Different Developmental Stages on Longissimus Dorsi to Identify Genes Underlying Intramuscular Fat Content in Wannanhua Pigs. Genes (Basel). 2023,14(4):903.

Reviewer 2 Report

Comments and Suggestions for Authors

This manuscript presents a comprehensive study combining proteomic and transcriptomic approaches to understand the molecular mechanisms underlying intramuscular fat (IMF) deposition in Wannanhua pigs, which is noted for its high IMF content, making it an ideal model for studying lipid deposition in pigs. The study aims to identify genes, proteins, and molecular pathways involved in intramuscular fat deposition.

This study addresses a significant gap in understanding the molecular basis of IMF deposition in pig skeletal muscle. It contributes to the field by identifying candidate genes and pathways that could improve meat quality traits in pig breeding. The research employs robust methods, including 4D label-free quantitative proteomic analysis and transcriptomic analysis. The use of parallel reaction monitoring (PRM) for validation of differentially expressed proteins (DEPs) adds credibility to the findings. The integration of transcriptomic and proteomic data provides a more holistic view of the molecular mechanisms involved. Findings from this study have practical implications for the breeding and genetic improvement of pigs, especially in enhancing meat quality traits.

This study identifies several candidate genes but does not delve into functional validation. Future studies could focus on experimental manipulation of these genes to confirm their roles in IMF deposition. While the study identifies key pathways, a more in-depth analysis of these pathways and their specific roles in IMF deposition would provide a clearer understanding.

Line 53-54: The Materials and Methods section states that pigs at 120 and 240 days of age were used in this study, rather than pigs at the same age with different intramuscular fat content. What is the rationale for this choice? The authors mentioned that "Our study of WH pigs at different developmental stages revealed the fastest muscle development and fat deposition between 120 d old and 240 d old." without any supporting material. Obviously, there must be time-dependent differentially expressed genes at these two time points in protein mass spectrometry and RNA sequencing analyses.

Line 84-95: Important information such as sex and genetic background (family line), body weight and backfat thickness at slaughter, etc. of the six pigs was not stated. Because there is some spatial and temporal variability in the expression of genes and proteins, these factors could affect the results of the experiments. Also, what was the intramuscular fat content of each of them? Please provide these important data.

Figure 1A: the font is so small that I can’t see clearly.

Please explain why the validation results for GAPDH in Result 3.5 are so unstable; we were under the impression that this was a housekeeping gene with stable expression.

In Figure 6, I suggest you swap the colors of up- and down-regulated proteins. Up-regulated proteins are shown in red, and down-regulated proteins are shown in green.

The layout of the images is poor, similar images are divided into 2 sections which is not conducive to the reader's overall understanding of the article.

Comments on the Quality of English Language

Line 16: “are” should be change to “is”

Please double check the full text for writing issues. If p-value appears as "P-values", "P-value", "P- values", "p < 0.05","P < 0.05", etc.

Author Response

1.Line 53-54: The Materials and Methods section states that pigs at 120 and 240 days of age were used in this study, rather than pigs at the same age with different intramuscular fat content. What is the rationale for this choice? The authors mentioned that "Our study of WH pigs at different developmental stages revealed the fastest muscle development and fat deposition between 120 d old and 240 d old." without any supporting material. Obviously, there must be time-dependent differentially expressed genes at these two time points in protein mass spectrometry and RNA sequencing analyses.

Reply:Thank you very much for your suggestion. The IMF content and the changes of muscle fiber types of WH pigs were measured at different developmental stages (LD60, LD120 and LD240). The results have been published in the following papers [Li X et al., 2023]. The present study was based on the transcriptomic analysis of the fastest stage of IMF deposition (LD120 and LD240 ) in WH pigs.

Li X, Yang Y, Li L, Ren M, Zhou M, Li S. Transcriptome Profiling of Different Developmental Stages on Longissimus Dorsi to Identify Genes Underlying Intramuscular Fat Content in Wannanhua Pigs. Genes (Basel). 2023,14(4):903.

2.Line 84-95: Important information such as sex and genetic background (family line), body weight and backfat thickness at slaughter, etc. of the six pigs was not stated. Because there is some spatial and temporal variability in the expression of genes and proteins, these factors could affect the results of the experiments. Also, what was the intramuscular fat content of each of them? Please provide these important data.

Reply:Thank you very much for your suggestion. It has been revised in the newly submitted paper. There were significant differences in IMF content between the two groups. The measurements have been published in a previous transcriptome data paper [Li X et al., 2023].

Li X, Yang Y, Li L, Ren M, Zhou M, Li S. Transcriptome Profiling of Different Developmental Stages on Longissimus Dorsi to Identify Genes Underlying Intramuscular Fat Content in Wannanhua Pigs. Genes (Basel). 2023,14(4):903.

  1. Figure 1A: the font is so small that I can’t see clearly.

 Reply: Thank you very much for your suggestion. It has been revised in the newly submitted paper.

  1. Please explain why the validation results for GAPDH in Result 3.5 are so unstable; we were under the impression that this was a housekeeping gene with stable expression.

 Reply: Thank you very much for your suggestion. Housekeeping, or reference genes (RGs) are, by definition, loci with stable expression profiles that are widely used as internal controls to normalize mRNA levels. But a large number of research results show that the vast majority of commonly used internal reference genes are not absolutely stable expression, but only relatively stable expression in cells and tissues under certain experimental conditions[Jung M et al., 2007; Strauss P et al., 2021]. The objects of this study were longissimus dorsi muscle tissues of the same breed at different developmental stages, so the expression of GAPDH was different.

Jung M, Ramankulov A, Roigas J, Johannsen M, Ringsdorf M, Kristiansen G, Jung K. In search of suitable reference genes for gene expression studies of human renal cell carcinoma by real-time PCR. BMC Mol Biol. 2007, 8:47.

Strauss P, Mikkelsen H, Furriol J. Variable expression of eighteen common housekeeping genes in human non-cancerous kidney biopsies. PLoS One. 2021, 16(12):e0259373.

  1. In Figure 6, I suggest you swap the colors of up- and down-regulated proteins. Up-regulated proteins are shown in red, and down-regulated proteins are shown in green.

Reply:Thank you very much for your suggestion. It has been revised in the newly submitted paper.

  1. The layout of the images is poor, similar images are divided into 2 sections which is not conducive to the reader's overall understanding of the article.

Reply:Thank you very much for your suggestion. It has been revised in the newly submitted paper.

7.Line 16: “are” should be change to “is”

 Reply: Thank you very much for your suggestion. It has been revised in the newly submitted paper.

8.Please double check the full text for writing issues. If p-value appears as "P-values", "P-value", "P- values", "p < 0.05","P < 0.05", etc.

Reply: Thank you very much for your suggestion. It has been revised in the newly submitted paper.

Round 2

Reviewer 1 Report

Comments and Suggestions for Authors

Most of the concerns were addressed. Thanks

Author Response

Most of the concerns were addressed. Thanks

Reply: Thank you very much for your suggestion.